# Functional Component Isolated from *Phaseolus vulgaris* Lectin Exerts In Vitro and In Vivo Anti-Tumor Activity Through Potentiation of Apoptosis and Immunomodulation

**DOI:** 10.3390/molecules26020498

**Published:** 2021-01-18

**Authors:** Peipei Wang, Xiaohong Leng, Jiaqi Duan, Yue Zhu, Jue Wang, Zirui Yan, Shitong Min, Dapeng Wei, Xia Wang

**Affiliations:** 1Department of Immunology, West China School of Basic Medical Sciences & Forensic Medicine, Sichuan University, Chengdu 610041, China; ppwangmail@163.com (P.W.); lengxiaohong1225@163.com (X.L.); wendee_tuan@163.com (J.D.); zzy.scu@163.com (Y.Z.); shitongminmail@163.com (S.M.); dwweimail@163.com (D.W.); 2State Key Laboratory of Biotherapy, West China Hospital, Sichuan University, Chengdu 610041, China; eternityjue@gmail.com (J.W.); ziruiyanyan@gmail.com (Z.Y.)

**Keywords:** PHA-L, functional component, anti-tumor activity, immunomodulation, toxicity

## Abstract

Phytohemagglutinin (PHA), the lectin purified from red kidney bean (*Phaseolus vulgaris*), is a well-known mitogen for human lymphocyte. Because it has obvious anti-proliferative and anti-tumor activity, PHA may serve as a potential antineoplastic drug in future cancer therapeutics. However, the literature is also replete with data on detrimental effects of PHA including oral toxicity, hemagglutinating activity, and immunogenicity. There is a critical need to evaluate the functional as well as the toxic components of PHAs to assist the rational designs of treatment with it. In this report, we performed SDS-PAGE to identify components of PHA-L, the tetrameric isomer of PHA with four identical L-type subunits, and then characterized biological function or toxicity of the major protein bands through in vitro experiments. It was found that the protein appearing as a 130 kD band in SDS-PAGE gel run under the condition of removal of β-mercaptoethanol from the sample buffer together with omission of a heating step could inhibit tumor cell growth and stimulate lymphocyte proliferation, while most of the 35 kD proteins are likely non-functional impurity proteins and 15 kD protein may be related to hemolytic effect. Importantly, the 130 kD functional protein exhibits promising in vivo anti-tumor activity in B16-F10 melanoma C57 BL/6 mouse models, which may be achieved through potentiation of apoptosis and immunomodulation. Altogether, our results suggest that PHA-L prepared from crude extracts of red kidney bean by standard strategies is a mixture of many ingredients, and a 130 kD protein of PHA-L was purified and identified as the major functional component. Our study may pave the way for PHA-L as a potential anticancer drug.

## 1. Introduction

Plant lectins are carbohydrate-binding proteins of non-immune origin ubiquitously distributed in a variety of plant species. During the middle of the 20th century, Nowell made the discovery that phytohemagglutinin (PHA), the lectin of the red kidney bean (*Phaseolus vulgaris*), is a mitogen, and plant lectins were brought into the limelight and there have been great strides in the knowledge of them [1,2]. Hundreds of plant lectins have been purified and their biochemical properties, carbohydrate-binding specificities and biological functions have been characterized in detail [3,4]. In addition to mitogenic simulation, it has been reported that plant lectins exhibit antifungal effects, HIV reverse transcriptase inhibition, and anti-proliferative and anti-tumor activity. Plant lectins have also been shown to have promising biological and medical uses for recognition of microorganisms, blood typing, isolation of glycoconjugates from cells, recognizing tools to differentiate malignant tumors from benign ones, and targeting of drugs to the gastrointestinal tract [5,6]. Therefore, the extensive exploration of plant lectins for novel biological action and potential health benefits is both timely and relevant.

*Phaseolus vulgaris* denotes the common bean, which includes kidney bean, pea bean, pink bean, black turtle bean, flageolet bean, white bean, yellow bean, and so on. They constitute one of the most important varieties of cultivated grain legumes, especially in tropical and subtropical countries, and are widely used for direct human consumption. PHA, the lectin purified from red kidney bean, is a kind of high molecular glycoprotein composed of galactose, *N*-acetylglucosamine, and mannose. In general, PHA is a tetramer built out of two types of polypeptide chains called L and E, reflecting their preferential binding to erythrocytes and leukocytes, respectively. Therefore, five possible tetrameric isomers (approximately 130 kD), namely E4, E3L1, E2L2, E1L3, and L4, can be randomly formed [7,8]. PHA-L, the L4 tetramer that consists of four identical L-type subunits, has been widely used as a mitogen to stimulate T lymphocyte proliferation in vitro. In recent years, the biomedical application of PHA-L in cancer diagnosis and treatment is gaining impetus. As aberrant glycosylation patterns of cancer cells have been shown to be relevant for tumor progression in different tumor entities, PHA-L, which specially recognizes β(1,6)-branched oligosaccharides, has been applied to detecting the glycosylation status of tumor cells to provide information on metastasis and prognosis [9]. More importantly, PHA-L has been reported to kill tumor cells through inducing apoptosis or autophagy both in cultured cells and in mice models, suggesting that PHA-L may serve as a potential antineoplastic drug in future cancer therapeutics [10].

Amazingly, although plant lectins were first described several decades ago, many questions about their potential biological effects remain obscure, especially the toxicity of lectin from kidney bean (PHA) [11]. Studies have shown that PHA is toxic to beetle larvae that eat cereal and legume seed [12]. Toxicity of PHA was also analyzed in a rodent model. Although the toxicity of PHA lectin to rats has not been observed, it has been found that the growth of animals is slow, and the nitrogen fixation capacity of intestinal mucosal cells is reduced [13]. In addition, PHA was found to affect lipid catabolism by reducing insulin level in plasma and strongly inhibiting hydrochloric acid secretion by rat gastric basal cells [14]. Recently, the ability of PHA to induce allergic reaction has also been reported, and the administration of PHA to rats was found to induce immunoglobulin (Ig) E-mediated reactions, and sometimes simultaneously with IgG-mediated reactions [15,16,17,18,19]. Thus, the toxicity and safety of PHA should be carefully evaluated in considering the application of it in therapeutics for humans. As the general methods for preparation of plant lectins include precipitation with salts or acids, followed by several chromatographic purification strategies, the identification of the actual active proteins from crude extracts has gained tremendous emphasis in recent years, which may provide a better understanding of both the beneficial and detrimental effects of lectins [20,21,22].

In this paper, we performed SDS-PAGE to identify protein bands for PHA-L, and then characterized biological functions or toxicity of the major bands. The anti-tumor activity of the proteins appearing as major bands was investigated in cultured tumor cell lines as well as in B16-F10 melanoma C57 BL/6 mouse models. To our knowledge, this is the first study addressing the separation and verifications of the functional components of PHA-L through both in vitro and in vivo experiments. We found that the 130 kD protein band appearing in SDS-PAGE gel run under the condition of removal of β-mercaptoethanol from the sample buffer together with omission of a heating step represents the major functional component of PHA-L, which could inhibit tumor cell growth and enhance anti-tumor immunity, while the 15 kD protein may be related to hemolytic effects. Our study may pave the way for PHA-L as a potential anticancer drug.

## 2. Results

### 2.1. Protein Profile of PHA-L

PHA-L was subjected to SDS-PAGE under two conditions: standard SDS-PAGE (inclusion of β-mercaptoethanol in the sample buffer with heating step) and modified SDS-PAGE (removal of β-mercaptoethanol from the sample buffer together with omission of a heating step). The gel was then stained with Coomassie brilliant blue as well as silver. Surprisingly, PHA-L showed four major protein bands at approximately 130, 35, 25, and 15 kD in modified SDS-PAGE gel and appeared as three major protein bands at approximately 35, 25, and 15 kD in standard SDS-PAGE gel (Figure 1). It is worth noting that the 130 kD protein band disappeared and the 35 kD protein band was significantly enhanced in standard SDS-PAGE gel compared with that in modified SDS-PAGE gel, whereas the 25 and 15 kD bands showed no obvious change. The results suggest that the 130 kD protein represents a tetramer of PHA-L that may dissociate into 35 kD monomer under standard denaturing condition.

### 2.2. Characterization of Functional Protein Bands from PHA-L

As PHA-L appeared as multiple bands in SDS-PAGE gel, we cut the protein bands from the gel, that is, 130, 35, and 15 kD bands in modified SDS-PAGE gel, and 35 kD bands in standard SDS-PAGE gel. Then, the gel slices were cut into small pieces and put into phosphate buffered saline (PBS) to let the protein be released from the gels. As PHA-L is known to be a potent lymphocyte mitogen, the mitogenic activity of released protein from each band was investigated by lymphocyte transformation experiment using human peripheral mononuclear cells. Surprisingly, the results showed that only the proteins released from 130 kD band in modified SDS-PAGE gel and 35 kD band in standard gel possessed definite mitogenic activity (Figure 2a–c), which strongly suggests that the 130 kD protein that appears in polyacrylamide gels under modified condition is the functional tetrameric component of PHA-L and could be dissociated into functional 35 kD monomer when β-mercaptoethanol was added into the sample buffer followed by a heating step. The result also suggests that most of the 35 kD protein that appears on gel run under modified condition is likely impurity without function, and the activity of PHA-L mainly exists in the 130 kD tetramer and its content is unexpectedly low among the whole reagent. Consistent to that, the mitogenic activity of 130 kD protein released from modified SDS-PAGE gel and 35 kD protein from denatured gel was significantly stronger than that of PHA-L (Figure 2a–c). We next investigated the erythroagglutination activity of PHA-L and released proteins from each band. As expected, PHA-L began to agglutinate a 2% suspension of human erythrocytes at 25 μg/mL, while proteins released from 130 kD and 35 kD bands both in modified and standard SDS-PAGE gels showed no visible erythroagglutination activity at 200 μg/mL. Interestingly, 15 kD protein caused partial hemolysis at 100 μg/mL (Figure 2d). Our results showed that the toxicity of PHA-L might be related to non-functional bands, such as the band with molecular weight of 15 kD.

### 2.3. In Vitro Anti-Tumor Effect of Functional Protein from PHA-L

As PHA-L has been previously reported to exhibit anti-tumor effect, we then tested the in vitro anti-tumor activity of proteins released from each band, that is, 130, 35, and 15 kD bands in modified SDS-PAGE gel, to identify the functional components of PHA-L with cytotoxicity to tumor cells. As shown in Figure 3, the protein of 130 kD could potently inhibited the in vitro proliferation of A549 (human non-small cell lung cancer cells), Jurkat (human acute T cell leukemia cells), and B16-F10 (mouse melanoma cells), as detected by CCK8 assay, with IC_50_ values of 14.48 ± 6.17 μg/mL (A549 cells), 0.03 ± 0.01 μg/mL (Jurkat cells), and 11.78 ± 2.17 μg/mL (B16-F10 cells), respectively. The anti-proliferation activity of 130 kD protein to tumor cells was more potent than that of PHA-L as the corresponding IC_50_ values for PHA-L were 41.53 ± 3.83 μg/mL (A549 cells), 0.63 ± 0.13 μg/mL (Jurkat cells), and 29.55 ± 3.92 μg/mL (B16-F10 cells), respectively. As expected, both the 35 kD protein and 15 kD protein showed marginal anti-proliferation effect on those three tumor cell lines. These results confirm the anti-tumor effect of PHA-L, and most importantly, prove that the major functional component of PHA-L with inhibitory effect on tumor cells is also the 130 kD protein.

### 2.4. In Vivo Anti-Tumor Effect of 130 kD Protein from PHA-L

The in vivo anti-tumor effect of 130 kD protein was validated with a melanoma syngeneic C57 BL/6 mouse model. As described in the Materials and Methods Section, B16-F10 cells were injected subcutaneously into the flank of C57 BL/6 mice for the development of tumors, and then the mice were treated with PBS control (NC), PHA-L, or 130 kD protein by intramuscular injection. In consideration of rather low content of 130 kD protein in total PHA-L, the dosage of PHA-L used in mice is 10 times that of 130 kD protein. As shown in Figure 4a, PHA-L (3 mg/kg) caused moderate retardation of tumor growth compared with that in the PBS-treated animals, whereas a pronounced tumor growth inhibition was seen in mice treated with 130 kD protein (0.3 mg/kg). The results of tumor weight assessed at the end of the experiment also supported that a much lower dose of 130 kD protein exhibits more potent antitumor activity than PHA-L, with the tumor inhibitory rate of 66% (130 kD protein-treated group) vs. 33% (PHA-L-treated group) (Figure 4b–d). To investigate whether the enhanced antitumor effect observed in 130 kD protein-treated animals is due to augmentation of apoptotic cell death, excised tumors were evaluated for apoptosis by TdT-mediated dUTP Nick-End Labeling (TUNEL) assay. As shown in Figure 4e,f, while only a few apoptotic cells (TUNEL-positive cells) were seen in the PHA-L-treated tumors, the number of apoptotic cells in the tumor tissues from animals treated with 130 kD protein was significantly increased. The in vivo anti-tumor effect of 130 kD protein was further evaluated through monitoring mouse survival using the same syngeneic melanoma mouse model. The mice were observed for about 40 days after inoculation of tumor cells, and the times of death for the mice were recorded. The survival curves showed that both PHA-L and 130 kD protein treatment could significantly prolong the survival time of mice compared with that of the PBS-treated group, in which 130 kD protein showed better effect of improving survival (Figure 5). These results provide strong evidence supporting that the 130 kD functional protein exhibits promising anti-tumor activity in vivo through potentiation of apoptosis.

### 2.5. In Vivo Immunoregulatory Activity of 130 kD Protein from PHA-L

Immunotherapy, that is, targeting the immune system, not the tumor itself, marks an entirely different way of treating cancer and has become an effective strategy for cancer treatment. As 130 kD protein from PHA-L exhibited in vitro mitogenic activity on human mononuclear cells, we then investigated whether the 130 kD protein has immunoregulatory activity in vivo, which may contribute to its in vivo anti-tumor effect in melanoma-bearing mice. Detection of CD3-positive lymphocytes and CD8-positive lymphocytes in formalin-fixed paraffin-embedded (FFPE) tumor tissues by immunohistochemistry (IHC) showed that both PHA-L and 130 kD protein treatment significantly promoted the infiltration of CD3-positive lymphocytes and CD8-positive lymphocytes in tumor tissue compared with that in the PBS control mice, and 130 kD protein exhibited a more prominent effect compared with PHA-L (Figure 6a). Consistent to that, both the granzyme B and perforin expression level, the markers of cytotoxicity of immunocytes, were found to be increased in tumor tissues of PHA-L-treated as well as 130 kD protein-treated mice compared with that in control mice (Figure 6b). To further confirm the in vivo immunoregulation effects of 130 kD protein, the spleen cells of mice in each group were prepared and stained by Wright’s staining. Both the spleen cells prepared from PHA-L-treated mice and 130 kD protein-treated mice showed significantly increased lymphocyte transformation rate compared with that of the spleen cells from the control mice (Figure 6c). In addition, their cytotoxicity to YAC-1 cells was also significantly enhanced compared with that of the spleen cells from the control group (Figure 6d). Altogether, these results suggest that the in vivo anti-tumor effect of 130 kD protein is likely achieved partly through upregulation of immune function.

## 3. Discussion

As a class of sugar-binding proteins of non-immune origin, plant lectins have been considered by most biomedical scientists for use in therapy in a variety of diseases, especially in the field of cancer [23,24,25,26]. Therefore, increasing concerns are currently aroused about the efficacy and safety of plant lectins. There is a critical need to evaluate the functional as well as the toxic components of plant lectins to assist the rational designs of treatment with lectins.

In this study, the components of PHA-L were firstly analyzed by SDS-PAGE. Surprisingly, it was found that PHA-L is a mixture composed of multiple components appearing as multiple bands from 130 to 15 kD in polyacrylamide gels. Nevertheless, it is reasonable as PHA-L is directly purified from crude extract of red kidney bean. The strategy commonly used for lectin purification from *Phaseolus vulgaris* beans is affinity chromatography using adsorbents of immobilized matrix-bound glycoproteins and glycopeptides after salt precipitation. Although there have been nearly 3000 articles reporting on the purification of *Phaseolus vulgaris* lectins as the main subject, the contamination of other isolectins, that is, L3E1, L2E2, L1E3, and E4, as well as other plant proteins, cannot be completely avoided.

The biological functions or toxicity of the major bands appearing in polyacrylamide gels were then characterized. The results showed that the 130 kD protein in modified SDS-PAGE gel is the major functional component with mitogenic activity, which retained the typical tetrameric structure of PHA-L. Under standard denaturing condition, the tetrameric 130 kD protein could dissociate into 35 kD monomer, which also showed moderate mitogenic activity after being released from gel. The results indicate that the functional PHA-L protein is quite stable as electrophoresis neither under modified nor standard condition causes a significant inhibitory effect on protein activity. Although modified SDS-PAGE was run under the condition of removal of β-mercaptoethanol from the sample buffer together with omission of a heating step, it cannot rule out the possibility that SDS in running buffer and in polyacrylamide gel could denature the proteins and turn the 130 kD tetramer into 35 kD monomer. However, marginal mitogenic activity of the 35 kD protein released from gel run under modified condition strongly indicate that most of them are likely impurity without function. It is noteworthy that the 15 kD protein may be the toxic component of PHA-L as it could cause partial hemolysis at a concentration of 100 μg/mL. As we know, this is the first report on the characterization of functional and toxic components of PHA-L, and our results showed that the toxicity of PHA-L might be related to non-functional bands, such as the band with molecular weight of 15 kD.

In addition to its mitogenic activity, the anticarcinogenic potential of *Phaseolus vulgaris* lectins has also attracted much attention in recent years. Although the underlying molecular mechanisms are still not fully clarified, numerous studies have shown the beneficial effects of *Phaseolus vulgaris* lectins in the possible therapies for lung cancer, liver cancer, breast cancer, melanoma, lymphoma, and so on [27,28,29,30]. In fact, as early as about 20 years ago, Ewen and colleagues reported that the inclusion of *Phaseolus vulgaris* lectin in diet can remarkably reduce the growth of murine non-Hodgkin lymphomas, either in an intra-peritoneal ascites tumor or a solid subcutaneous tumor mouse model [31,32]. Recently, Kochubei et al. reported that PHA and its isolectins were shown to induce the apoptosis in human HEp-2 carcinoma cells via increasing pro-apoptotic protein Bax and activating caspases-3 [33]. Consistently, our study showed that the 130 kD protein is the functional component of PHA-L with cytotoxicity to tumor cells, as demonstrated by in vitro as well as in vivo experiments. Importantly, treatment with 130 kD protein could significantly prolong the survival time of B16-F10 cells-inoculated mice. Furthermore, detection of apoptosis of tumor tissues by TUNEL assay proved that the 130 kD functional protein exhibits anti-tumor activity in vivo, at least partially through potentiation of apoptosis.

Given the fact that PHA-L has dual properties (cytotoxicity and immunomodulation) [34], it will be intriguing if the 130 kD protein can also exert immunomodulatory activity in vivo. As expected, the infiltration of both CD3-positive lymphocytes and CD8-positive lymphocytes, and the expression level of the granzyme B and perforin in tumor tissue were significantly enhanced in 130 kD protein-treated mice. It is generally known that after CD8^+^ T cells recognize the antigen, degranulation leads to the release of perforin, which promotes the formation of pores on the tumor cell membrane, allowing granzyme B to enter the cytoplasm to activate the pro-apoptotic Bcl-2 family members [35,36,37,38]. In addition, spleen cells from 130 kD protein-treated mice showed increased lymphocyte transformation rate, and cytotoxicity to YAC-1 cells. These data provided strong evidence supporting that 130 kD protein could play an immunomodulatory role in vivo, which may contribute to its antitumor effect in the mice model. A previous study from other researcher also showed that *Phaseolus vulgaris* lectins could simultaneously activate the immune system by secreting various interleukins, including interleukin-2, tumor necrosis factor alpha, and interferon-gamma, which were involved in induction of apoptosis of the tumor cells [28]. Therefore, the dual properties of *Phaseolus vulgaris* lectins make them good potential candidates for tumor therapy.

In summary, although for the past several decades plant lectins were considered as toxic substances to cells and animals, there is a current tendency in the field of lectin research to shift the use of plant lectins from detection to treatment due to the cytotoxic-, apoptosis-, and autophagy-inducing effects and immunomodulation activity of plant lectins. However, compared with the studies on the anticarcinogenic potential of other lectin families, such as Con A, mistletoe lectins, and so on, studies on *Phaseolus vulgaris* lectins are still limited [39,40,41]. The potential of *Phaseolus vulgaris* lectins ought to be elucidated on the molecular level as well as in clinical trials. Other than that, more cost-effective novel purification methods as well as the production of lectin with recombinant techniques need to be designed to get the lectin with high yields, purity, and activity to meet the increasing requirements for exploring its biological and biomedical applications.

## 4. Materials and Methods

### 4.1. Reagents and Cells

PHA-L (L2769) was purchased from Sigma (Saint Louis, MO, USA). PAGE Gel Silver Staining Kit (G7210) was purchased from Solarbio (Beijing, China). Cell Counting Kit-8 (CK04) was from Dojindo (Kumamoto, Japan), and CytoTox 96 Non-radioactive Cytotoxicity Assay (G1782) was from Promega (Madison, WI, USA). In Situ Cell Death Detection Kit-TMR red was from Roche Applied Science (12156792910, Mannheim, Germany). Protein ladder was purchased from Thermo (26616), (Vilnius, Lithuania). DAB substrate kit was from ZSBIO (ZLI-9017, Beijing, China), and hematoxylin was from Solarbio (G4070). All the antibodies against CD3E (512415), CD8A (510793), Granzyme B (252579), and Perforin (500093) were purchased from ZEN BIO (Chengdu, China). The A549 (human non-small cell lung cancer cells), Jurkat (human acute T cell leukemia cells), B16-F10 (mouse melanoma cells) and YAC-1 (mouse lymphoma cells) cell lines were from American type culture collection (ATCC).

### 4.2. SDS-PAGE and Silver Staining

For SDS-PAGE under non-denaturing condition, the sample was prepared by adding 20 μg of PHA-L into loading buffer with β-mercaptoethanol followed by heating. For SDS-PAGE under denaturing condition, the sample was prepared by adding equal amount of PHA-L into loading buffer without β-mercaptoethanol and heating. The samples were resolved by 10% SDS-PAGE. Then, the gel was subjected to Coomassie brilliant blue R-250 staining and silver staining to visualize protein bands in the gel.

### 4.3. Acquisition of Different Band Proteins of PHA-L

Forty mg of PHA-L was subjected to SDS-PAGE electrophoresis under modified condition as well as standard condition, and 130, 35, and 15 kD bands in modified SDS-PAGE gel and a 35 kD band in standard gel were cut according to the marker position, respectively. The gel slices were cut into small pieces and incubated with PBS overnight to let the protein be released from the gels. The gel mixture was then centrifuged to get rid of gel. The supernatant liquid was collected into a dialysis bag for dialysis with PBS to remove small-molecule compounds that may also be released from gel, as they may be toxic to cells or mice. After overnight dialysis, the proteins were concentrated by using centrifugal filters from Millipore and then the volume of retrieved samples was adjusted to 2 mL with PBS. The protein solution was filtered through a 0.2-micron filter before lyophilization. The proteins were reconstituted using sterile distilled water before use. The existence and purity of isolated proteins from gel were validated by running SDS-PAGE under condition without heating and β-mercaptoethanol treatment (data not shown). The concentration of purified proteins was determined by using Pierce Rapid Gold BCA Protein Assay Kit according to the manufacturer’s instructions. The purification yield from cut gel is about 2.8 mg (130 kD protein), 3.9 mg (35 kD protein), and 8.0 mg (15 kD protein), respectively.

### 4.4. Lymphocyte Proliferation Assay and Wright’s Staining

The human peripheral blood mononuclear cells were isolated and seeded in 96-well plates followed by treatment with 20 μg/mL of PHA-L or component proteins with different molecular weight, as indicated for 72 h, respectively. The cell proliferation was then detected by CCK8 assay according to the manufacturer’s instructions (Dojido, Kumamoto, Japan). Briefly, CCK8 reagent was added into the wells (10 μL/well) and incubated for 4 h, and the absorbance of the samples was measured at 450 nm using a microplate reader. All the experiments were repeated three to five times and the average was shown. The percentage of viable cells was calculated using the following formula: cell viability (%) = (absorbance of treated sample/absorbance of control) × 100. In addition, the cell smears were stained with Wright’s stain for 2 min, followed by adding equal amount of phosphate buffer solution into Wright’s dye solution, mixed, and incubated for another 5 min. After washing away the dye solution and drying, the smear slides were evaluated under microscope and cells with typical transformed lymphocyte morphology were quantified by counting a total of 300 cells in five fields (200×). The spleen cells from mice of each group were also stained with Wright’s dye solution and quantified in the same way.

### 4.5. Hemagglutination Assay

Erythroagglutination activity of PHA-L and its major components with different molecular weights was measured by hemagglutination assay. In brief, 2% suspension of human erythrocytes was loaded in a 96-well microtiter U-plate. Then, different doses of PHA-L or major component proteins (from 200 to 12.5 μg/mL) were added into the wells and incubated for 30 min at 37 °C. Hemagglutination was observed by the presence of agglutinated red blood cells in the well.

### 4.6. Tumor Cell Culture and Cell Viability Assay

The human non-small cell lung cancer cells A549, human acute T cell leukemia cells Jurkat, and mouse melanoma cells B16-F10 were purchased from American Type Culture Collection (ATCC, Manassas, VA, USA). The cells were cultured in RPMI 1640 supplemented with 10% fetal bovine serum (FBS), 1 mM of glutamine, 100 U/mL of penicillin, and 100 μg/mL of streptomycin. The cells were treated with different doses of PHA-L, or proteins released from 130, 35, and 15 kD bands in modified SDS-PAGE gel for 72 h. Cell viability was assessed using a CCK8 kit. The percentage of viable cells was calculated using the following formula: cell viability (%) = (absorbance of treated sample/absorbance of control) × 100.

### 4.7. Cell-Mediated Cytotoxicity Assay

A cytotoxicity assay based on the release of lactate dehydrogenase (LDH) was conducted using CytoTox 96 non-radioactive cytotoxicity detection kit from Promega according to the manufacturer’s instructions. Briefly, the spleen cells (effector cells) from mice of each group were co-cultured with YAC-1 cells (target cells) in an effector and target cell ratio of 10:1 overnight. For the preparation of target cell maximum LDH release, 10 µL of the Lysis Solution (10×) per 100 µL of culture medium was added into the target cell control and incubated for 45 min prior to harvesting the supernatants, which should yield complete lysis of target cells. 50 µL aliquots from all wells were transferred to a fresh 96-well flat-bottom (enzymatic assay) plate. LDH activity was determined by adding equal volumes of reaction mixture to each well and incubating for up to 30 min. The absorbance of the samples was measured at 490 nm using a plate reader. Cell death was calculated using the following formula:% Cytotoxicity = ((Experimental − Effector Spontaneous − Target Spontaneous)/(Target Maximum − Target Spontaneous)) × 100.(1)

### 4.8. Murine Tumor Models

For syngeneic allograft models, 6-week-old C57 BL/6 mice were injected subcutaneously with B16-F10 (1 × 10^6^ cells) into the flank for the development of tumors. All procedures involving animals and their care were conducted in accordance with the guidelines of the Institutional Animal Care and Use Committee of Sichuan University. On the second day after cell inoculation, the mice were randomly divided into three subgroups and administrated with PBS control, PHA-L (3 mg/kg), or 130 kD protein (0.3 mg/kg) by intramuscular injection. Dosing was carried out every day for five consecutive days, stopped for two days, and then continued for another five consecutive days. After palpable tumors had developed, tumor was measured with a clipper every day, and tumor volume (V) was calculated using the following formula: V = length × width^2^/2. On day 16 post-tumor implantation, animals were euthanized and sacrificed. Excised tumors were weighed. The anti-tumor effects of PHA-L and 130 kD protein were expressed as the tumor inhibitory rate: ((mean tumor weight of PBS group—tumor weights of drug-treated group)/mean tumor weight of PBS group) × 100%. In addition, the survival of mice treated with PHA-L or 130 kD protein were also compared with that of mice treated with PBS control using the same mouse model and treatment protocol. The mice were observed for about 40 days, and the death time of mice was recorded.

### 4.9. TUNEL Assay and IHC

To detect cell death in tumor tissue, tumor samples from each group were harvested and in situ TUNEL assays were carried out in paraffin-embedded tumor tissue sections. TUNEL assay was based on labeling of DNA strand breaks by using the in situ cell death detection kit according to the manufacturer’s instructions. Briefly, after dewaxation, rehydration, protease treatment, and permeabilization, the tumor tissue sections were incubated with the terminal deoxynucleotidy transferase labeling reaction mixture for 60 min at 37 °C. Nucleus was stained with DAPI. The sections were evaluated under a laser scanning confocal microscope using an excitation wavelength of 543 nm for TUNEL staining and 488 nm for DAPI staining. The apoptotic (TUNEL-positive) cells showed red fluorescence. To observe tumor immune cell infiltration, IHC staining was carried out on FFPE tumor tissue sections with different primary antibodies and a Signal Stain DAB substrate kit, as described previously [42]. Antibodies against CD3E, CD8A, Granzyme B, and Perforin were used to detect the expression of corresponding proteins followed by nuclear staining with hematoxylin. Image quantification was performed using Image J, and each data point is the mean of three independent sum intensity measurements taken from representative fields.

### 4.10. Statistics

All quantitative data are expressed as mean ± SD or ± SEM. The survival curve was drawn by GraphPad Prism. Statistical significance was also examined by GraphPad Prism, and *p* < 0.05 was considered statistically significant.

## Figures and Tables

**Figure 1 molecules-26-00498-f001:**
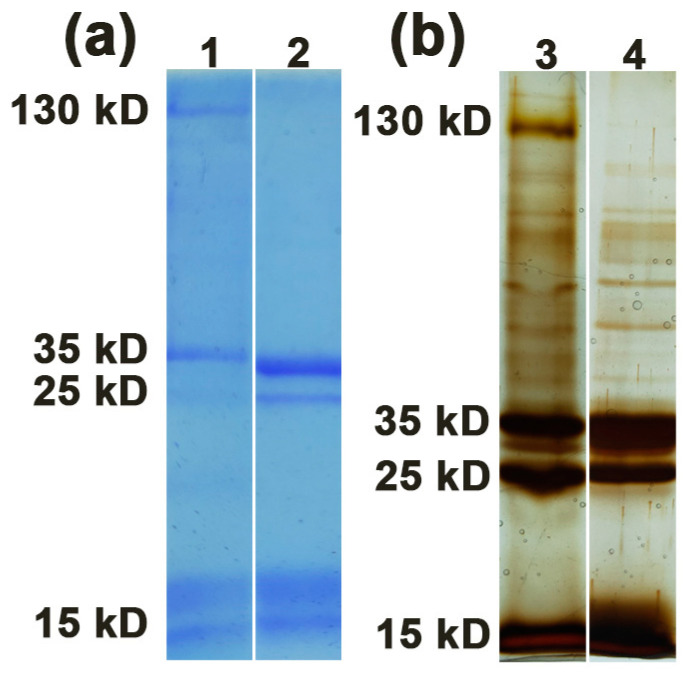
Protein profile of PHA-L by SDS-PAGE under modified condition (lane 1 and 3) and standard condition (lane 2 and 4). (**a**) SDS-PAGE gel was stained with Coomassie brilliant blue; (**b**) SDS-PAGE gel was stained by silver staining.

**Figure 2 molecules-26-00498-f002:**
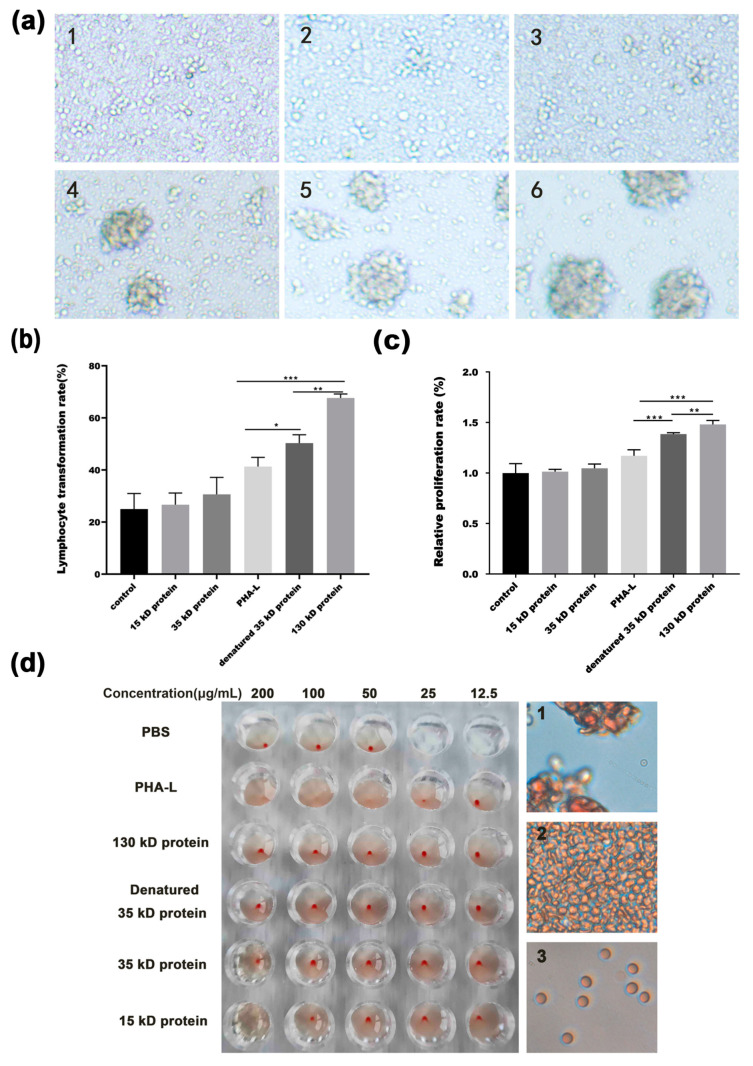
The characterization of mitogenic activity and erythroagglutination activity of major proteins from PHA-L. (**a**) Representative pictures of human peripheral mononuclear cells treated with PHA-L or proteins released from major bands for 72 h, as indicated (1. PBS control, 2. 15 kD protein, 3. 35 kD protein, 4. PHA-L, 5. 35 kD protein in standard denatured gel, 6. 130 kD protein); (**b**) Lymphocyte transformation rate detected by Wright’s stain. Columns, mean of three experiments; bars, SD. * *p* < 0.05, ** *p* < 0.01, *** *p* < 0.001; (**c**) Lymphocyte proliferation rate detected by Cell Counting Kit-8 (CCK8) assay. Columns, mean of three experiments; bars, SD. ** *p* < 0.01, *** *p* < 0.001; (**d**) 2% suspensions of human erythrocytes were treated with PHA-L or proteins released from major bands, as indicated for 30 min at 37 °C. Left, the red dot in the pore indicated no agglutination, and the liquid mixed indicated agglutination or hemolysis. Right, representative pictures were taken under microscope (1. agglutination state, 2. non-agglutination state, 3. hemolytic state).

**Figure 3 molecules-26-00498-f003:**
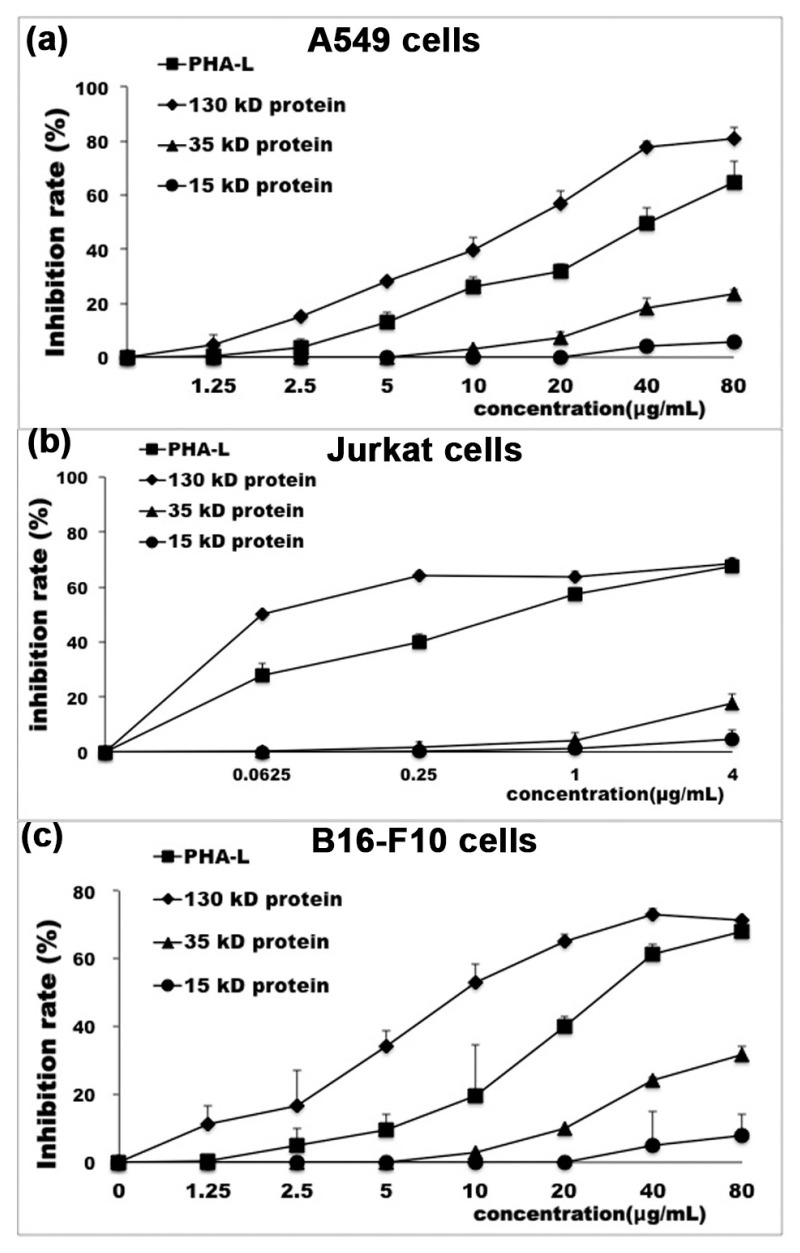
Dose response curve of PHA-L and proteins released from major bands on tumor cell proliferation. (**a**) The proliferation inhibition rate of A549 treated with different doses of PHA-L, and proteins released from 130, 35, and 15 kD bands in modified SDS-PAGE gel, as indicated, was detected by CCK8. 130 kD protein, IC_50_ = 14.48 ± 6.17 μg/mL; PHA-L, IC_50_ = 41.53 ± 3.83 μg/mL; (**b**) The proliferation inhibition rate of Jurkat cells was detected as in A549 cells. 130 kD protein, IC_50_ = 0.03 ± 0.01 μg/mL; PHA-L, IC_50_ = 0.63 ± 0.13 μg/mL; (**c**) The proliferation inhibition rate of B16-F10 cells was detected as in A549 cells. 130 kD protein, IC_50_ = 11.78 ± 2.17 μg/mL; PHA-L, IC_50_ = 29.55 ± 3.92 μg/mL.

**Figure 4 molecules-26-00498-f004:**
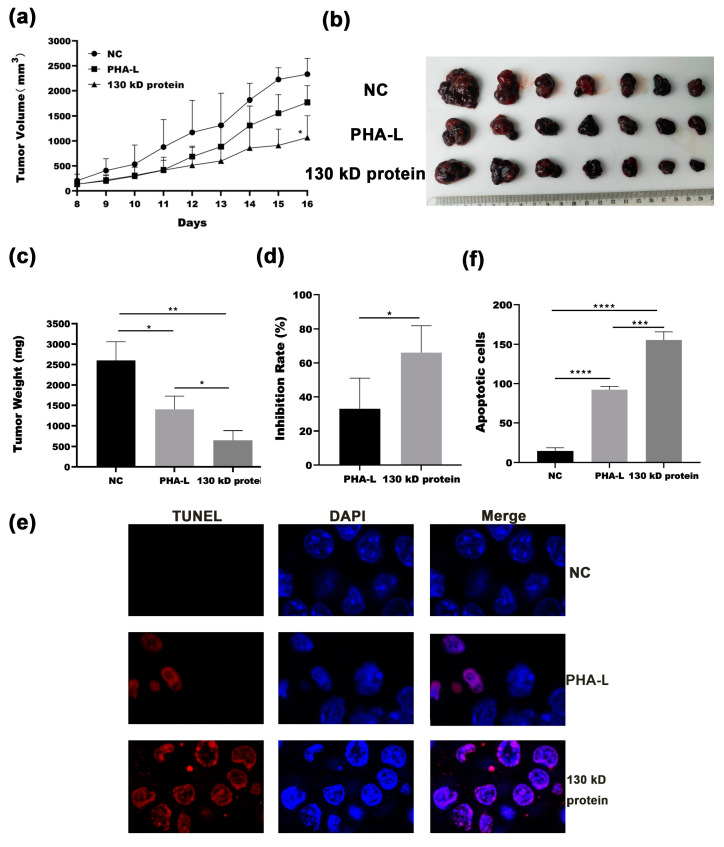
Anti-tumor effect of 130 kD protein in vivo. (**a**) C57 BL/6 mice were injected (subcutaneously) with B16-F10 cells. The tumor-bearing mice from each group (*n* = 7 mice/group) were then treated with PBS, PHA-L (3 mg/kg), or 130 kD protein (0.3 mg/kg) intramuscularly. The volume of tumors was measured daily from day 8 over the study period, and the mean tumor volume of each group was shown. NC vs. PHA-L, ns, * NC vs. 130 kD protein, *p* < 0.05; (**b**) Sixteen days after tumor cell inoculation, tumors were removed from mice and observed; (**c**) The mean weight of tumors in each group was presented. Columns represent mean in each group, bars represent SD. * *p* < 0.05, ** *p* < 0.01; (**d**) Tumor inhibition rates were calculated and shown. Columns represent mean in each group, bars represent SD. * *p* < 0.05; (**e**) Sections of tumor tissue were subjected to apoptosis analysis by TUNEL assay. Nucleus was stained with DAPI. Samples were analyzed under a laser scanning confocal microscope using an excitation wavelength of 543 nm for TUNEL staining, and 488 nm for DAPI staining. Representative images for each group were shown (1000×); (**f**) Apoptosis (TUNEL-positive) cells (red) were quantified by counting in a total of 300 cells in five fields. Average apoptotic cell numbers per 300 cells were shown as mean ± SD. Columns represent mean in each group, bars represent SD. *** *p* < 0.001, **** *p* < 0.0001.

**Figure 5 molecules-26-00498-f005:**
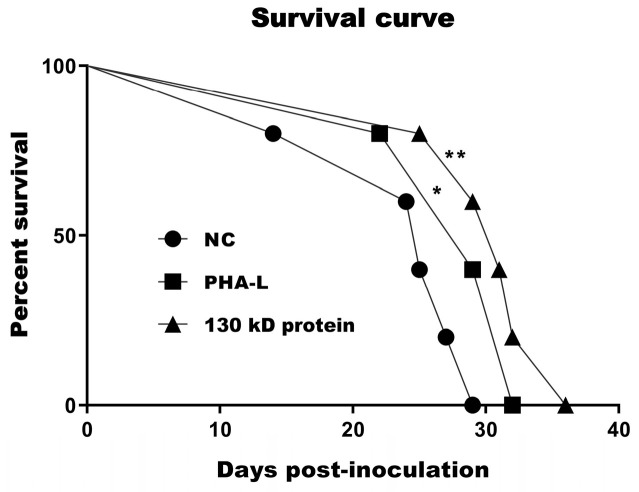
130 kD protein could prolong the survival time of mice compared with that of PBS-treated mice. * NC vs. PHA-L, *p* < 0.05, ** NC vs. 130 kD protein, *p* < 0.01.

**Figure 6 molecules-26-00498-f006:**
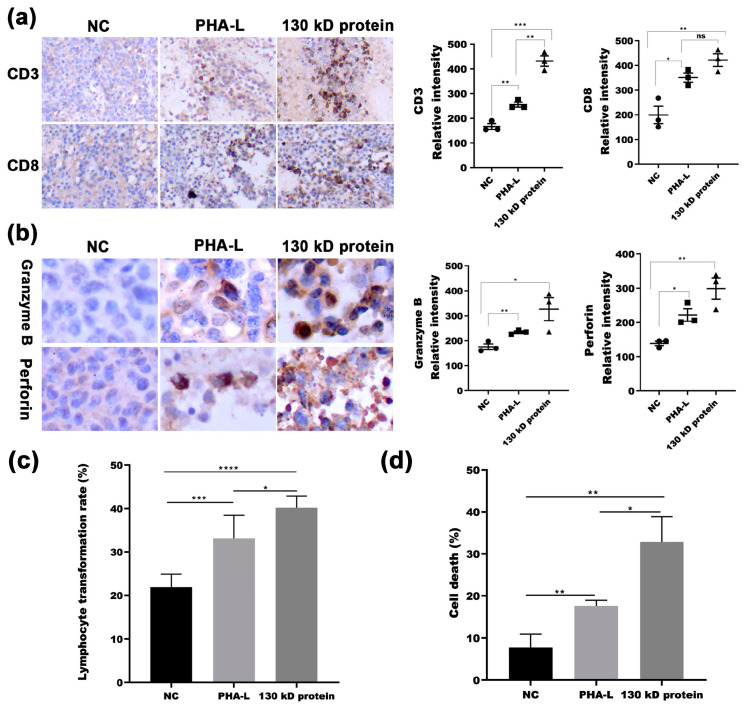
In vivo immunoregulatory activity of 130 kD protein. (**a**) Representative images of immunohistochemical staining for mouse CD3 and CD8 in FFPE sections of tumor tissues (100×). Quantitation of image intensity for CD3 (left) and CD8 (right) were shown. The mean relative intensity value was displayed above the plots, bars represent SEM. (**b**) Quantitation of images of immunohistochemical staining for granzyme B and perforin in FFPE sections of tumor tissues (400×). Quantitation of image intensity for granzyme B (left) and perforin (right) were shown. The mean relative intensity value was displayed above the plots, bars represent SEM. (**c**) Lymphocyte transformation rate of spleen cells of mice from each group was calculated after Wright’s stain. (**d**) Cell death of YAC-1 co-incubated with spleen cells of mice from each group was detected by CytoTox 96 non-radioactive cytotoxicity detection kit (LDH assay). * *p* < 0.05, ** *p* < 0.01, *** *p* < 0.001, **** *p* < 0.0001.

## Data Availability

The data presented in this study are available in this article.

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
