# Peer review of "Functional Component Isolated from Phaseolus vulgaris Lectin Exerts In Vitro and In Vivo Anti-Tumor Activity through Potentiation of Apoptosis and Immunomodulation"

_molecules, 2021, doi:10.3390/molecules26020498_

Round 1

Reviewer 1 Report

The manuscript by Wang et al. that focuses on anti-tumor activity of lectin isolated from P. vulgaris was significantly improved. However, the writing still needs some improvement/additional revision. 

In the new version the authors still fail to perform clear-cut electrophoresis experiments. While the is no problem with SDS-PAGE in denaturing conditions, the methodology and results of “non-denatured SDS-PAGE gel” still require additional clarification.

First of all, the “non-denatured SDS-PAGE gel” is incorrect and most likely the authors have in mind “non-denaturing gel”. Secondly, as the authors acknowledged, SDS is a detergent that can cause proteins to denature. Therefore, despite the fact that the sample used for electrophoresis is not treated with a reducing agent and heated, it enters a gel that has a detergent that most likely denatures part of the protein. This experiment (native electrophoresis, native PAGE) should be performed without SDS present in a sample buffer, gel, as well as the buffer used to run the gel.

In summary, the manuscript should be revised and all text fragment where the authors refer to “non-denatured SDS-PAGE”, “SDS-PAGE under non-denaturing condition”, etc. should be corrected and clarified. Especially confusing and incorrect is the part of Material and Methods section (paragraph 4.2), in which the authors explain the difference between “non-denaturing” and “denaturing” conditions.

There are also some other issues that need to be clarified.

  1. Paragraph 4.7 (last sentence)

How protein was quantified after the overnight dialysis and sterile filtration?

  1. Experimental errors should be reported for all IC50 values mentioned in the manuscript.
  2. Please clarify/correct the statement on sample availability (line 528).

Reviewer 2 Report

This manuscript reported the protein band composition of phytohemagglutinin (PHA) and the anti-tumor function of the different protein bands.

The authors have added detailed methods on how 130kD protein was released from the gel and used for in-vitro and in-vivo treatment. They used dialysis, filtration and lyophilisation to isolate the 130kD protein from cut non-denature gels. After those isolation procedures, the authors should run non-denature gel again to validate the existence and purity of 130kD protein in the solution, which is ready to be used for treatments. In addition, methods on how the concentration of purified 130kD protein was determined, the purification yield from cut gel and dilution details were still missing.

Writing can be improved, it would be better to move some discussion text from the results section to discussion section.

Round 2

Reviewer 1 Report

All my previous comments were addressed, and I do not have any new comments.

Reviewer 2 Report

Questions have been adequately addressed. I do not have further comments.

This manuscript is a resubmission of an earlier submission. The following is a list of the peer review reports and author responses from that submission.

Round 1

Reviewer 1 Report

〇Comments

This manuscript by Peipei Wang et al., addresses a separation and verifications of the functional bands of PHAL in vitro and in vivo. The authors founds that the hemolytic effect and anti-tumor effect differed between 15 kDa and 130 kDa of PHAL. It is useful to clear the activity of functional proteins for anti-cancer drugs super efficacy and low toxicity. However, the results are not well discussed. The results should be improved with more explanations and the text needs to be modified more correctly. I hope you to help with my comments.

  1. L95-; The authors should show what you explain definitely. e.g. ‘the three bands’ is about ‘15, 35 and 130 kDa’, ‘with an IC50 of 6.95 mM’ showed the inhibition with ‘130 kDa of PHAL’ in text.
  2. L109-112, L263-272; The same sentence is used at plural points. Please check it.
  3. In this study, the authors showed the 130 kDa protein is more effectively than total PHAL in the various activity and curative effect. Why is the reason? (e.g. is the reason that PHAL have a conformational problem or an influence of inactive domain?)
  4. Legume lectins are built up of protomers of approximately 30 kDa and PHAL is composed of tetramer. What kind of proteins is 15 and 25 kDa. Additionally, how do the authors think about the relation between each protein band and glycan binding site (or recognition of glycans) of PHAL?
  5. Uniform each unit, e.g. 4 h vs. 72 hours; 20 minutes vs. 30 min; 400 μg/ml vs. 20 ug/ml; 50 μL vs. 10 ul. Please check all text.

Reviewer 2 Report

This manuscript reported the protein band composition of phytohemagglutinin (PHA) and the anti-tumor function of the different protein bands. The authors performed SDS-PAGE to identify protein bands for L-subunit of PHA and conducted in-vitro and in-vivo experiments in melanoma to test the function of the protein bands. The authors concluded that the 130kD protein band at non-denatured state inhibited tumor cell growth and enhanced anti-tumor immunity, while the 15kD protein band may be related to the hemolytic effects.

My major comment is:

For treatment of cells or mice with protein bands, the authors briefly wrote in the method section that “protein bands were added” or “by intramuscular injection every day”. Did the authors cut the protein band from the SDS-PAGE and then add the gel slice to cell culture medium or inject the gel slice to mice? How did the authors know that the protein was released from the gel slice and  interacted with the cells? Was the gel itself toxic to cells or mice? Was the gel slice sterile? A majority of the findings in this manuscript rely on this method, however, it is not convincing that adding gel slice to cell culture or injection of the gel slice to mice will work appropriately to test the function of the protein band.

Reviewer 3 Report

The manuscript by Wang et al. focuses on the effect of lectin isolated from P. vulgaris on melanoma cells. While the topic is interesting, unfortunately Abstract, Introduction, Results and some fragments of Discussion sections are very poorly written, and currently the manuscript cannot be recommended for publication. All mentioned sections have to be significantly re-written/corrected.

In addition, the authors treat SDS-PAGE without heat treatment and beta-mercaptoethanol as non-denaturing, which is not true. SDS is a detergent that can cause proteins to denature.